# SampleFix: Learning to Correct Programs by Efficient Sampling of Diverse Fixes

**Hossein Hajipour**[1], **Apratim Bhattacharya** [2], **Mario Fritz**[1]

[1]CISPA Helmholtz Center for Information Security
[2] Max Planck Institute for Informatics
`{hossein.hajipour,fritz}@cispa.saarland,`
`abhattac@mpi-inf.mpg.de`

## Abstract

Automatic program correction holds the potential of dramatically improving the productivity of programmers. Recent advances in machine learning and NLP have rekindled the hope to eventually fully automate the process of repairing programs. A key challenge is ambiguity, as multiple codes – or fixes – can implement the same functionality, and there is uncertainty on the intention of the programmer. As a consequence, datasets by nature fail to capture the full variance introduced by such ambiguities. Therefore, we propose a deep generative model to automatically correct programming errors by learning a *distribution* over potential fixes. Our model is formulated as a deep conditional variational autoencoder that can efficiently sample diverse fixes for a given erroneous program. In order to account for inherent ambiguity and lack of representative datasets, we propose a novel regularizer to encourage the model to generate diverse fixes. Our evaluations on common programming errors show strong improvements over the state-of-the-art approaches. Extended version available at `https://arxiv.org/abs/1906.10502` .

## 1 Introduction

Software development is a time-consuming and expensive process. Unfortunately, programs are written by humans typically come with bugs. Debugging is also typically performed by humans and can contain mistakes. Therefore, automatically locating and correcting program errors offers the potentials to increase productivity as well as improve the correctness of software.

Advances in deep learning [10–12], computer vision [6, 13] and NLP [2, 15] has dramatically boosted the machine's ability to automatically learn representations of natural data such as images and language contents. These advances in deep learning and the advent of large corpora of source code [1] provide new opportunities

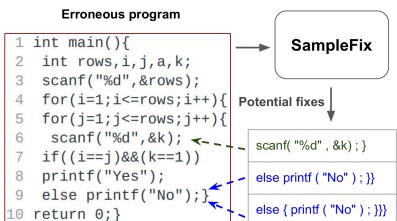

Figure 1: SampleFix captures the inherent ambiguity of the possible fixes by sampling multiple potential fixes for the given erroneous real-world program. Potential fixes with the same functionality are highlighted with the same color.

toward harnessing deep learning methods to understand, generate, or debug programs. There has been an increasing interest to use deep learning to tackle the "common programming errors" (compile errors), which includes missing delimiters, extraneous symbols, and type errors. Gupta et al. [8] propose DeepFix, a sequence-to-sequence model to predict a fix for the erroneous program. RLAssist [7] repairs the programs by employing reinforcement learning, and Yasunaga and Liang [17] propose DrRepair to resolve the syntax error by introducing a program feedback graph. In these works, the problem is formulated as a deterministic task, where the model is trained to predict a single location and fix for each error. However, different codes – and their fixes – can express the same functionality. Besides, there is also uncertainty about the intention of the programmer (Figure 1).

NeurIPS 2020 Workshop on Computer-Assisted Programming, Vancouver, Canada.

Hence, we propose a generative framework to automatically correct programming errors by learning the distribution of potential fixes. We investigate different solutions to model the distribution of the fixes and sample multiple fixes, including different variants of Conditional Variation Autoencoders (CVAE) and beam search decoding. It turns out CVAE and beam search decoding are complementary (section 3), while CVAE is computationally more efficient in comparison to beam search decoding. Figure 1 shows an example of the generated diverse and correct fixes using our proposed approach. To summarize, the contributions of this paper are as follows, 1. We propose an efficient generative method to automatically correct common programming errors. 2. We propose a novel regularizer to encourage the model to generate diverse fixes. 3. Our generative model together with the diversity regularizer shows strong improvement over the state-of-the-art approaches.

## 2 SampleFix: Generative Model for Code Fixes

Repairing the common program errors is a challenging task due to the ambiguity in potential corrections. Therefore, we propose a deep generative framework to automatically localize and repair errors by learning the distribution of potential fixes – rather than a single fix – given the erroneous program. We address the problem of generating multiple fixes using CVAE and beam search decoding.

In the CVAE formulation, we model the distribution of the fixes using the conditional latent variable. The distribution of latent variables is learned using pairs of erroneous programs and the corresponding fixes. Furthermore, to encourage diversity in the fixes, we propose a novel regularizer that encourages the model to generate diverse samples by maximizing the distance among drawn fixes.

In the inference time given an erroneous program, our model draws $T$ candidate fixes from the learned distribution. To select one out of $T$ candidates, we employ a compiler. The compiler evaluates each fix by compiling the updated program. Among the fixes, we select the fix which resolves the largest number of error messages. To resolve the remaining error(s), we follow Gupta et al. [8] and iteratively input the updated program to the model. In the following, we provide detail of our approach.

### 2.1 Conditional Variational Autoencoders for Generating Fixes

CVAEs [14] model conditional distributions $p_\theta(\mathbf{y}|\mathbf{x})$ using latent variables $\mathbf{z}$. The conditioning introduced through $\mathbf{z}$ enables the modelling of complex multi-modal distributions. As powerful transformations can be learned using neural networks, $\mathbf{z}$ itself can have a simple distribution which allows for efficient sampling. This model allows for sampling from $p_\theta(\mathbf{y}|\mathbf{x})$ given an input sequence $\mathbf{x}$, by first sampling latent variables $\hat{\mathbf{z}}$ from the prior distribution $p(\mathbf{z})$. During training, amortized variational inference is used and the latent variables $\mathbf{z}$ are learned using a recognition network $q_\phi(\mathbf{z}|\mathbf{x}, \mathbf{y})$, parametrized by $\phi$. In detail, the variational lower bound of the model is maximized,

$$\log(p(\mathbf{y}|\mathbf{x})) \geq \mathbb{E}_{q_\phi(\mathbf{z}|\mathbf{x},\mathbf{y})} \log(p_\theta(\mathbf{y}|\mathbf{z}, \mathbf{x})) - D_{\mathrm{KL}}(q_\phi(\mathbf{z}|\mathbf{x}, \mathbf{y}) \,||\, p(\mathbf{z}|\mathbf{x})). \tag{1}$$

Penalizing the divergence of $q_\phi(\mathbf{z}|\mathbf{x}, \mathbf{y})$ to the prior in Equation 1 allows for sampling from the prior $p(\mathbf{z})$ during inference. In practice, the variational lower bound is estimated as follow [14],

$$\hat{\mathcal{L}}_{\mathrm{CVAE}} = \frac{1}{T} \sum_{i=1}^{T} \log(p_\theta(\mathbf{y}|\hat{\mathbf{z}}_i, \mathbf{x})) - D_{\mathrm{KL}}(q_\phi(\mathbf{z}|\mathbf{x}, \mathbf{y}) \,||\, p(\mathbf{z}|\mathbf{x})) \;. \tag{2}$$

where, $\hat{\mathbf{z}}_i \sim q_\phi(\mathbf{z}|\mathbf{x}, \mathbf{y})$, and $T$ is the number of samples. In our model, the input $\mathbf{x}$ is the erroneous program and $\mathbf{y}$ is the fix that contains line of the error with the repaired line.

### 2.2 Enabling Diverse Samples using a Best of Many Objective

Casting our model in the CVAE framework would enable us to sample a set of candidate fixes for a given erroneous program. However, the standard variational lower bound objective does not encourage diversity in the candidate fixes [14]. In detail, as the average likelihood is considered, all candidate fixes must explain the "true" fix in training set well. To encourage diversity, we employ "Best of Many Samples" (BMS) objective proposed by Bhattacharyya et al. [3],

$$\hat{\mathcal{L}}_{\mathrm{BMS}} = \max_i \left( \log(p_\theta(\mathbf{y}|\hat{\mathbf{z}}_i, \mathbf{x})) \right) - D_{\mathrm{KL}}(q_\phi(\mathbf{z}|\mathbf{x}, \mathbf{y}) \,||\, p(\mathbf{z}|\mathbf{x})) \;. \tag{3}$$

Compared to Equation 2, this objective (Equation 3) encourages diversity in the model by considering highly likely candidate fixes. This enables the model to generate diverse and accurate candidates [3].

## 2.3 DS-SampleFix: Encouraging Diversity with a Diversity-sensitive Regularizer

To increase diversity using Equation 3 , we need to use around $T = 10$ samples during training. This is computationally prohibitive especially for large models, as it requires 10 times the memory or 10 times the number of forward passes. If $T$ is decreased, the objective behaves similarly to the CVAE objective. Therefore, in order to encourage the model to generate diverse fixes even when high values of $T$ are computationally prohibitive, we propose a novel regularizer that explicitly encourages the two closest candidate fixes to have maximum distance (Equation 4). This penalizes generating similar candidate fixes for a given erroneous program and thus encourages diversity in the set of candidate fixes. In comparison to Equation 3, we observe considerable gains with the use of only $T = 2$ samples,

$$\hat{\mathcal{L}}_{\text{DS-BMS}} = \max_i \left( \log(p_\theta(\mathbf{y}|\hat{\mathbf{z}}_i, \mathbf{x})) \right) + \min_{i,j} d(\hat{\mathbf{y}}^i, \hat{\mathbf{y}}^j) - D_{\text{KL}}(q_\phi(\mathbf{z}|\mathbf{x}, \mathbf{y}) \,||\, p(\mathbf{z}|\mathbf{x})) \ . \tag{4}$$

Note that the samples $\left\{ \hat{\mathbf{y}}^i, \hat{\mathbf{y}}^j \right\}$ can be of different lengths. Therefore, we pad the samples to equalize lengths. In practice, we find that the Euclidean distance performs best for the metric $d$ in Equation 4.

## 2.4 Beam Search Decoding for Generating Fixes

Beam search decoding is classical model to generate multiple outputs from a sequence-to-sequence model [16, 5]. In our generative model, we employ the beam search decoding to sample more diverse fixes. In detail, to sample multiple fixes we decode with beam width of size $K$ for each sample $\mathbf{z}$.

## 2.5 Model Architecture and Implementation Details

To ensure a fair comparison, our generative model is based on the sequence-to-sequence architecture, similar to Gupta et al. [8]. Figure 2 shows the architecture of our approach in detail. Note that the recognition network is available to encode the fixes to latent variables $\mathbf{z}$ only during training. All of the networks in our framework consists of 4-layers of LSTM cells with 300 units. The network is optimized using Adam optimizer [9] with the default setting. We use $T = 2$ samples to train our models, and $T = 100$ samples during inference.

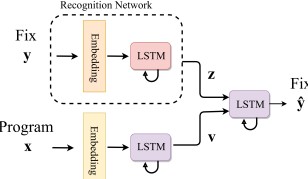

Figure 2: Overview of network architecture.

During inference, the conditioning erroneous program $\mathbf{x}$ is input to the encoder, which encodes the program to the vector $\mathbf{v}$. To generate multiple fixes using our decoder, the code vector $\mathbf{v}$ along with a sample of $\mathbf{z}$ from the prior $p(\mathbf{z})$ is input to the decoder. For simplicity, we use a standard Gaussian $\mathcal{N}(0, \mathbf{I})$ prior. The decoder is unrolled in time and output logits ($p_\theta(\mathbf{y}|\hat{\mathbf{z}}_i, \mathbf{x})$). To resolve multiple errors we use the iterative repair strategy as introduced by Gupta et al. [8].

# 3 Experiments

We evaluate our approach on the task of repairing common programming errors and compare it with the state-of-the-art approaches.

## 3.1 Dataset

We use the dataset published by Gupta et al. [8] as it's sizable and includes real-world data. It contains C programs written by students in an introductory programming course. The programs were collected using a web-based system [4]. These programs contain typographic and missing variable declaration errors. The dataset contains two sets of data, which are called synthetic and real-world data. The synthetic data contains the erroneous programs which are synthesized by mutating correct programs. The real-world data contains 6975 erroneous programs with 16766 error messages written by students. We use synthetic data to train and validate our model, and we use real-world data to test our model.

| Models | Typo | | Miss Dec | | All | | Speed (s) |
|---|---|---|---|---|---|---|---|
| | ✔ | 🐞 | ✔ | 🐞 | ✔ | 🐞 | |
| DeepFix [8] | 23.3% | 30.8% | 10.1% | 12.9% | 33.4% | 40.8% | - |
| *RLAssist* [7] | *26.6%* | *39.7%* | - | - | - | - | - |
| DrRepair [17] | - | - | - | - | 34.0% | - | - |
| Our DeepFix+ BS | 25.8% | 38.9% | 16.8% | 35.3% | 39.0% | 56.9% | 4.82 |
| Our SampleFix | 24.8% | 38.8% | 16.1% | 22.8% | 40.9% | 56.3% | 0.88 |
| Our DS-SampleFix | 27.7% | 40.9% | 16.7% | 24.7% | 44.4% | 61.0% | 0.88 |
| Our DS-SampleFix + BS | **27.8%** | **45.6%** | **19.2%** | **47.9%** | **45.2%** | **65.2%** | 1.17 |

Table 1: Results of performance comparison of DeepFix, RLAssist, DrRepair, DeepFix+ BS (beam search), SampleFix , DS-SampleFix, and DS-SampleFix + BS (beam search). Typo, Miss Dec, and All refer to typographic, missing variable declarations, and all of the error messages respectively. Speed denotes the computational time for sampling 100 fixes. ✔denotes completely fixed programs, while 🐞 refers to resolved error messages.

## 3.2   Evaluation

We evaluate our approach on real-world set of erroneous programs. In the real-world data, we don't have access to the intended fix(es). However, we can check the correctness of the program using the compiler. In Table 1 we compare our approaches with the state-of-the-art approaches. In this table (Table 1) we show the performance of DeepFix+ BS (beam search), SampleFix, and our DS-SampleFix. Furthermore, we show that DS-SampleFix can still take advantage of beam search. In order to do that, for each sample $z$ we decode with beam width of size 5, and to sample 100 fixes we draw 20 samples from $p(z)$. We also provide the sampling speed in terms of sampling 100 fixes for a given program using an average over 100 runs.

Table 1 shows that our approaches outperform DeepFix [8], RLAssist [7], and DrRepair [17] in resolving the error messages. This shows that generating multiple diverse fixes can lead to substantial improvement in performance. SampleFix, DS-SampleFix, and DS-SampleFix + BS (beam search) resolve 56.3%, 61.0%, and 65.2% of the error messages respectively. Furthermore, the performance advantage of DS-SampleFix over SampleFix shows the effectiveness of our novel regularizer.

Note that the concurrent work DrRepair [17] has achieved further improvements by relying on the compiler. While utilizing the compiler output seems to be beneficial, it also limits the generality of the approach. For a fair comparison, we report the performance of DrRepair without the compiler output, but consider informing our model by the compiler output an interesting avenue of future work.

**Qualitative Example.**   We illustrate diverse fixes generated by our DS-SampleFix in Figure 3 using an example with typographic errors with the corresponding drawn outputs of DS-SampleFix. In this example, there is a missing closing curly bracket after line 12. We can see that DS-SampleFix generates multiple correct fixes to resolve the error in the given program. This indicates that our approach is capable of handling inherent ambiguity and uncertainty in predicting fixes. More examples can be found in Appendix A.

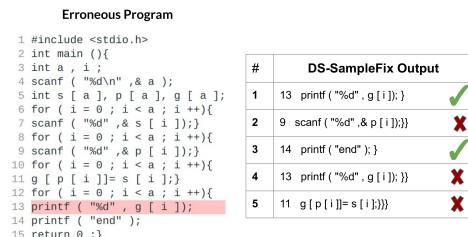

Figure 3: DS-SampleFix generates multiple correct fixes (line number with the corresponding fix). The green tick indicates the correct fix.

## 4   Conclusion

We propose a novel approach to correct common programming errors. We recognize and model the inherent ambiguity and uncertainty with predicting multiple fixes. In contrast to previous works, our novel approach is able to learn the distribution over candidate fixes rather than the most likely fix. We propose a novel diversity-sensitive regularizer that helps us overcome the inherent limitations of supervised datasets. We show that our approach is capable of generating multiple correct fixes. Furthermore, our evaluations show improvements over state-of-the-art approaches.

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

# Appendix A    Additional Qualitative Examples

Here we provide additional examples of repairing erroneous programs using our proposed approach. Figure 4 and 5 show examples of the generated multiple correct fixes using DS-SampleFix. In the code example given in Figure 4, there is a missing closing curly bracket. In this example DS-SampleFix generates four correct fixes for the given erroneous program. Among these fixes, the third and fourth fixes express the same functionality. In the code example given in Figure 5, there is an extraneous closing curly bracket. In this example, we can see DS-SampleFix generates two correct fixes that express different functionalities. Note that the model outputs a program location along with the corresponding fix.

Figure 6 and 7 show code examples where DS-SampleFix repairs multiple typographic and missing variable declaration errors. In Figure 6, we can see that DS-SampleFix resolves two typographic errors at line 10 and 16. In Figure 7, DS-SampleFix resolves missing variable declaration errors by defining variables *n*, *eof*, and *m*.

**Erroneous Program**

```
 1 #include <stdio.h>
 2 #include <stdlib.h>
 3 int main (){
 4 int i , j , b , h , k ;
 5 scanf ( "%d %d" ,& h ,& b );
 6 for ( i = 1 ; i <= h ; i ++) { k = i ;
 7 for ( j = 1 ; j <= b ; j ++){
 8 if ( j <( h - 1 - i )|| j >( b - i - h ))
 9 printf ( " " );
10 else {
11 printf ( "%d" ,( k % 10 ));
12 k ++;}
13 printf ( "\n" );}
14 return 0 ;}
```

| # | DS-SampleFix Output | |
|---|---|---|
| 1 | 12   k ++; }} | ✔️ |
| 2 | 9    printf ( " " ); } | ❌ |
| 3 | 11   printf ( "%d" ,( k % 10 )); } | ✔️ |
| 4 | 11   printf ( "%d" , k % 10 ); } | ✔️ |
| 5 | 13   printf ( "\n" ); }} | ✔️ |

Figure 4: Diverse fixes are generated by our approach (DS-SampleFix). The potential location of the error is highlighted. The green tick indicates the correct fix.

**Erroneous Program**

```
 1 #include <stdio.h>
 2 int main (){
 3 int i , j , l , k , temp , n , r ;
 4 scanf ( "%d %d \n" ,& n ,& k );
 5 int a [ n ];
 6 for ( i = 0 ; i < n ; i ++){
 7 scanf ( "%d" ,& a [ i ]);}
 8 for ( j = 0 ; j < n ; j ++){
 9 int min = j ;
10 for ( r = j + 1 ; r < n ; r ++){
11 if ( a [ r ]< a [ min ]){
12 min = r ;
13 temp = a [ j ];
14 a [ j ]= a [ min ];
15 a [ min ]= temp ;}}
16 printf ( "%d" , a [ j ]);}}
17 return 0 ;}
```

| # | DS-SampleFix Output | |
|---|---|---|
| 1 | 15   a [ min ]= temp ; } | ✔️ |
| 2 | 15   a [ min ]= temp ; }}} | ❌ |
| 3 | 16   printf ( "%d" , a [ j ]); } | ✔️ |
| 4 | 13   temp = a [ j ]; } | ❌ |
| 5 | 16   printf ( "%d" , a [ j ]); }} | ❌ |

Figure 5: Diverse fixes are generated by our approach (DS-SampleFix). The potential location of the error is highlighted. The green tick indicates the correct fix.

Figures 8 and 9 show two examples of resolving multiple errors using itrerative repair strategy, here we show 5 samples out of 100 drawn samples. In Figure 8a we can see that in the first iteration our approach resolves a typographic error at line 9, and in the second iteration (Figure 8b), after updating the program, it resolves another typographic error at line 11. Figure 9 show an example of iteratively resolving multiple missing variable declaration errors. Figures 9a and 9b show resolving multiple errors by defining variables *i* and *j* respectively. Note that the outputs contain only *_eos_* refer to the empty string.

**Erroneous Program**

```
1  #include <stdio.h>
2  int main (){
3  int m , n , t , j , sum [ 200 ], f [ 200 ], s ;
4  scanf ( "%d%d" ,& n ,& m );
5  int a [ 200 ];
6  for ( i = 1 ; i <= n ; i ++){
7  for ( j = 0 ; j <= m ; j ++){
8  scanf ( "%d" ,& f [ j ]);
9  sum [ i ]= 0 ;
10 sum [ i ]= sum [ i ]+ f [ j ];}}
11 x = sum [ i ];
12 for ( i = 2 ; i <= m ; i ++){
13 if ( x < sum [ i ])
14 x = sum [ i ];}
15 printf ( "%d" , x );}
16 scanf ( "%d" ,&)
17 return 0 ;}
```

**Repaired Program**

```
1  #include <stdio.h>
2  int main (){
3  int m , n , t , j , sum [ 200 ], f [ 200 ], s ;
4  scanf ( "%d%d" ,& n ,& m );
5  int a [ 200 ];
6  for ( i = 1 ; i <= n ; i ++){
7  for ( j = 0 ; j <= m ; j ++){
8  scanf ( "%d" ,& f [ j ]);
9  sum [ i ]= 0 ;
10 sum [ i ]= sum [ i ]+ f [ j ];}
11 x = sum [ i ];
12 for ( i = 2 ; i <= m ; i ++){
13 if ( x < sum [ i ])
14 x = sum [ i ];}
15 printf ( "%d" , x );}
16 scanf ( "%d" );
17 return 0 ;}
```

Figure 6: Example of resolving typographic errors using our approach (DS-SampleFix).

**Erroneous Program**

```
1  #include <stdio.h>
2  int max ( s [ int a ], s [ int b ]){
3  if ( s [ a ]> s [ b ])
4  return s [ a ];
5  else
6  return s [ b ];}
7  int main (){

8  int a ;
9  int char s [ n ];
10 int i = 1 ;
11 scanf ( "%d%d" ,& n ,& a );
12 while ( a != eof && i < n ){
13 s [ i ]= a ;
14 m = max ( s [ i ], s [ i - 1 ]);
15 i = i + 1 ;
16 scanf ( "%d" ,& a );}
17 printf ( "%d" , m );
18 return 0 ;}
```

**Repaired Program**

```
1  #include <stdio.h>
2  int max ( s [ int a ], s [ int b ]){
3  if ( s [ a ]> s [ b ])
4  return s [ a ];
5  else
6  return s [ b ];}
7  int main (){
8  int n ;
9  int eof ;
10 int m ;
11 int a ;
12 int char s [ n ];
13 int i = 1 ;
14 scanf ( "%d%d" ,& n ,& a );
15 while ( a != eof && i < n ){
16 s [ i ]= a ;
17 m = max ( s [ i ], s [ i - 1 ]);
18 i = i + 1 ;
19 scanf ( "%d" ,& a );}
20 printf ( "%d" , m );
21 return 0 ;}
```

Figure 7: Example of resolving missing variable declaration errors using our approach (DS-SampleFix).

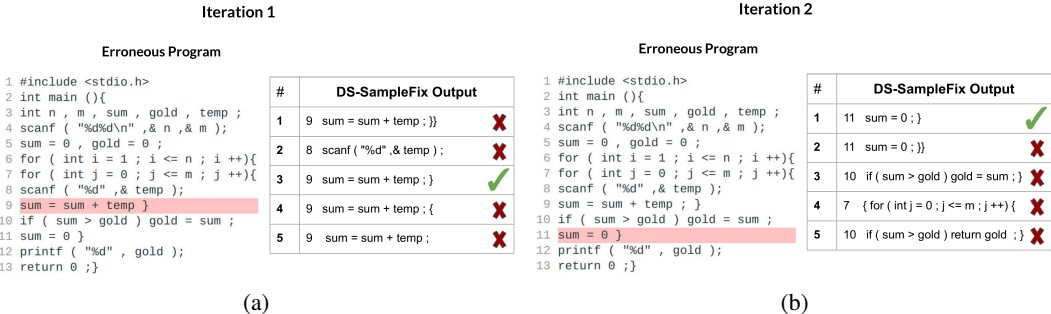

Figure 8: Example of iteratively resolving multiple typographic errors using our approach (DS-SampleFix).

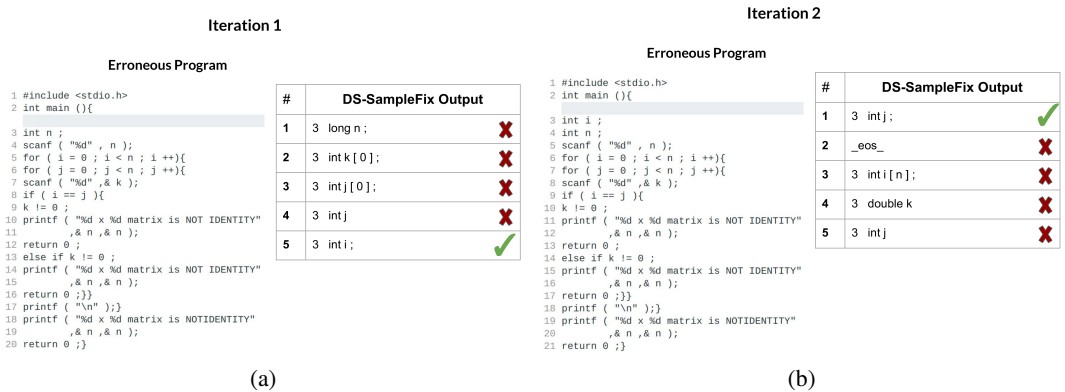

Figure 9: Example of iteratively resolving multiple missing variable declaration errors using our approach (DS-SampleFix).

