# OpenReview forum: "SampleFix: Learning to Correct Programs by Efficient Sampling of Diverse Fixes"
_NeurIPS.cc/2020/Workshop/CAP — NeurIPS 2020 CAP Workshop_

### Official Review · AnonReviewer1 · 2020-10-19

**Rating:** 5
**Confidence:** 4

**Review:**

### Summary ###
The paper addresses the problem of bug fixing using a conditional VAE that generates possible fixes, and a compiler that checks each of the suggested fixes. The paper also proposes a regularizer loss to control the diversity of suggested fixes.

Since the main contribution of the paper is its empirical results, I put more weight on the validity of the evaluation. I was not convinced that the used dataset is meaningful and challenging, that the right baselines were used, and that the comparison was fair for the baselines. I thus vote for rejection at this time. I hope that the authors will improve their evaluation and submit this as a full paper later.

### Strengths ###
The proposed loss function of taking the two candidates whose distance is the largest (Equation 4) is interesting.

### Weaknesses ###
The paper claims "strong improvements over state-of-the-art", but I am not sure that the evaluation is correct and fair.
For example, did the baselines get to use the same beam size?
Did all baselines get to use the same number of suggestions (T) and compiler checks?
Did all baselines have access to a compiler?
Since the model is relatively simple (LSTM seq2seq) - did all baselines get the same number of layers / LSTM units?
How does a simple LSTM seq2seq+attention+copy or a Transformer perform?

The dataset -
The example fixes are mostly trivial to solve.  Figures 3-4-5-6 just miss or have an extra closing curly bracket. Also, I am not sure that all the solutions that are marked as "correct" with green checkmarks are indeed "correct", as they are very different. For example, in Figure 5, why an assignment that is crucial for the correctness of the program can be correctly replaced with a `printf`?.
Figures 7,9 complete missing variable initializations, Figures 8(a) and Figure 8(b) fix missing semicolons. None of these examples really require machine learning to solve.

### Minor concerns ###
I am also bothered by the claims of the paper.
The authors claim that in previous work "model is trained to predict a single location and fix for each error", and that, in contrast, the proposed model learns a "distribution over potential fixes".
Don't all neural models can output a distribution over outputs, and sample or provide multiple candidates?

The authors also claim that their approach can "efficiently sample diverse fixes" - how is this approach more **efficient** than sampling from any other model?

I did not understand how the proposed BMS objective encourages diversity (Equation 3)

---

### Decision · Program_Chairs · 2020-11-03

**Decision:**

Accept

**Comment:**

I agree with some of the criticisms made by the reviewer.
In particular, the empirical evaluation could be substantially improved.
I would not accept this paper (or a longer version) to a major machine learning conference.
However, the paper does have merits and could in the future be improved into something more up to the standards of such a conference.
For a workshop, I think this level of polish is acceptable, and so am recommending acceptance.